# LIFELONG GENERATIVE MODELING

## ABSTRACT

Lifelong learning is the problem of learning multiple consecutive tasks in a sequential manner where knowledge gained from previous tasks is retained and used for future learning. It is essential towards the development of intelligent machines that can adapt to their surroundings. In this work we focus on a lifelong learning approach to generative modeling where we continuously incorporate newly observed streaming distributions into our learnt model. We do so through a student-teacher architecture which allows us to learn and preserve all the distributions seen so far without the need to retain the past data nor the past models. Through the introduction of a novel cross-model regularizer, the student model leverages the information learnt by the teacher, which acts as a summary of everything seen till now. The regularizer has the additional benefit of reducing the effect of catastrophic interference that appears when we learn over streaming data. We demonstrate its efficacy on streaming distributions as well as its ability to learn a common latent representation across a complex transfer learning scenario.

## 1 INTRODUCTION

Deep unsupervised generative learning allows us to take advantage of the massive amount of unlabeled data available in order to build models that efficiently compress and learn an approximation of the true data distribution. It has numerous applications such as image denoising, inpainting, super-resolution, structured prediction, clustering, pre-training and many more. However, something that is lacking in the modern ML toolbox is an efficient way to learn these deep generative models in a sequential, lifelong setting.

In a lot of real world scenarios we observe distributions sequentially. Examples of this include streaming data from sensors such as cameras and microphones or other similar time series data. A system can also be resource limited wherein all of the past data or learnt models cannot be stored. We are interested in the lifelong learning setting for generative models where data arrives sequentially in a stream and where the storage of all data is infeasible. Within the stream, instances are generated according to some non-observed distribution which changes at given time-points. We assume we know the time points at which the transitions occur and whether the latent distribution is a completely new one or one that has been observed before. We do not however know the underlying identity of the individual distributions. Our goal is to learn a generative model that can summarize all the distributions seen so far in the stream. We give an example of such a setting in figure 1(a) using MNIST LeCun & Cortes (2010), where we have three unique distributions and one that is repeated.

Since we only observe one distribution at a time we need to develop a strategy of retaining the previously learnt knowledge (i.e. the previously learnt distributions) and integrate it into future learning. To accumulate additional distributions in the current generative model we utilize a student-teacher architecture similar to that in distillation methods Hinton et al. (2015); Furlanello et al. (2016). The teacher contains a summary of all past distributions and is used to augment the data used to train the student model. The student model thus receives data samples from the currently observable distribution as well as synthetic data samples from previous distributions. This allows the student model to learn a distribution that summarizes the current as well as all previously observed distributions. Once a new distribution shift occurs the existing teacher model is discarded, the student becomes the teacher and a new student is instantiated.

We further leverage the generative model of the teacher by introducing a regularizer in the learning objective function of the student that brings the posterior distribution of the latter close to that of the

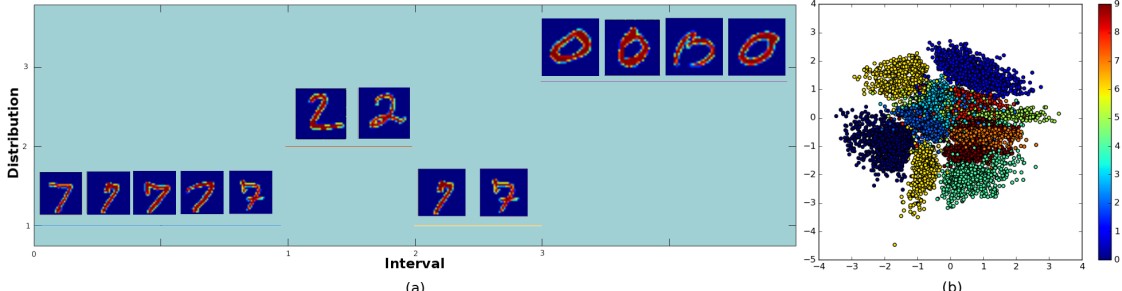

Figure 1: (a) Our problem setting where we sequentially observe samples from multiple unknown distributions; (b)Visualization of a learnt two-dimensional posterior of MNIST, evaluated with samples from the full test set.

former. This allows us to build upon and extend the teacher's generative model in the student each time the latter is re-instantiated (rather than re-learning it from scratch). By coupling this regularizer with a weight transfer from the teacher to the student we also allow for faster convergence of the student model. We empirically show that the regularizer allows us to learn a much larger set of distributions without catastrophic interference McCloskey & Cohen (1989).

We build our lifelong generative models over Variational Autoencoders (VAEs) Kingma & Welling (2014). VAEs learn the posterior distribution of a latent variable model using an encoder network; they generate data by sampling from a prior and decoding the sample through a conditional distribution learnt by a decoder network.

Using a vanilla VAE as a teacher to generate synthetic data for the student is problematic due to a couple of limitations of the VAE generative process. 1) Sampling the prior can select a point in the latent space that is in between two separate distributions, causing generation of unrealistic synthetic data and eventually leading to loss of previously learnt distributions. 2) Additionally, data points mapped to the posterior that are further away from the prior mean will be sampled less frequently resulting in an unbalanced sampling of the constituent distributions. Both limitations can be understood by visually inspecting the learnt posterior distribution of a standard VAE evaluated on test images from MNIST as shown in figure 1(b). To address the VAE's sampling limitations we decompose the latent variable vector into a continuous and a discrete component. The discrete component is used to summarize the discriminative information of the individual generative distributions while the continuous caters for the remaining sample variability. By independently sampling the discrete and continuous components we preserve the distributional boundaries and circumvent the two problems above.

This sampling strategy, combined with the proposed regularizer allows us to learn and remember all the individual distributions observed in the past. In addition we are also able to generate samples from any of the past distributions at will; we call this property consistent sampling.

## 2 RELATED WORK

Past work in sequential learning of generative models has focused on learning Gaussian mixture models Singer & Warmuth (1999); Declercq & Piater (2008) or on variational methods such as Variational EM Ghahramani & Attias (2000). Work that is closer to ours is the online or sequential learning of generative models in a streaming setting. Variational methods have been adapted for a streaming setting, e.g: Streaming Variational Bayes Broderick et al. (2013), Streaming Variational Mixture models Tank et al. (2015), and the Population Posterior McInerney et al. (2015). However their learning objectives are very different from ours. The objective of these methods is to adjust the learnt model such that it reflects the current data distribution as accurately as possible, while forgetting the previously observed distributions. Instead we want to do lifelong learning and retain all previously observed distributions within our learnt model. As far as we know our work is the first one that tries to bring generative models, and in particular VAEs, into a lifelong setting where distributions are seen, learnt, and remembered sequentially.

VAEs rely on an encoder and a decoder neural network in order to learn the parameters of the posterior and likelihood. One of the central problems that arise when training a neural network in an sequential manner is that it causes the model to run into the problem of catastrophic interference McCloskey & Cohen (1989). Catastrophic interference appears when we train neural networks in a sequential manner and model parameters start to become biased to the most recent samples observed, while forgetting what was learnt from older samples. This generally happens when we stop exposing the model to past data. There have been a number of attempts to solve the problem of catastrophic interference in neural networks. These range from distillation methods such as the original method Hinton et al. (2015) and ALTM Furlanello et al. (2016), to utilizing privileged information Lopez-Paz et al. (2016), as well as transfer learning approaches such as Learning Without Forgetting Li & Hoiem (2016) and methods that relay information from previously learnt hidden layers such as in Progressive Neural Networks Rusu et al. (2016) and Deep Block-Modular Neural Networks Terekhov et al. (2015). All of these methods necessitate the storage of previous models or data; our method does not.

The recent work of elastic weight consolidation (EWC) Kirkpatrick et al. (2017) utilizes the Fisher Information matrix (FIM) to avoid the problem of catastrophic interference. The FIM captures the sensitivity of the log-likelihood with respect to the model parameters; EWC leverages this (via a linear approximation of the FIM) to control the change of model parameter values between varying distributions. Intuitively, important parameters should not have their values changed, while non-important parameters are left unconstrained. Since EWC assumes model parameters being distributed under an exponential family, it allows for the utilization of the FIM as a quadratic approximationJeffreys (1946) to the Kullback-Leibler (KL) divergence. Our model makes no such distributional assumptions about the model parameters. Instead of constraining the parameters of the model as in EWC, we restrict the posterior representation of the student model to be close to that of the teacher for the previous distributions accumulated by the teacher. This allows the model parameters to vary as necessary in order to best fit the data.

## 3 BACKGROUND

We consider an unsupervised setting where we observe a sample $\mathbf{X}$ of $K \geq 1$ realizations $\mathbf{X} = \{\mathbf{x}^{(0)}, \mathbf{x}^{(1)}, ..., \mathbf{x}^{(K)}\}$ from an unknown true distribution $P^*(\mathbf{x})$ with $\mathbf{x} \in \mathcal{R}^N$. We assume that the data is generated by a random process involving a non-observed random variable $\boldsymbol{z} \in \mathcal{R}^M$. In order to incorporate our prior knowledge we posit a prior $P(\boldsymbol{z})$ over $\boldsymbol{z}$. Our objective is to approximate the true underlying data distribution by a model $P_{\boldsymbol{\theta}}(\mathbf{x})$ such that $P_{\boldsymbol{\theta}}(\mathbf{x}) \approx P^*(\mathbf{x})$.

Given a latent variable model $P_{\boldsymbol{\theta}}(\mathbf{x}|\boldsymbol{z})P(\mathbf{z})$ we obtain the marginal likelihood $P_{\boldsymbol{\theta}}(\mathbf{x})$ by integrating out the latent variable $\mathbf{z}$ from the joint distribution. The joint distribution can in turn be factorized using the conditional distribution $P_{\boldsymbol{\theta}}(\mathbf{x}|\boldsymbol{z})$ or the posterior $P_{\boldsymbol{\theta}}(\boldsymbol{z}|\mathbf{x})$.

$$P_{\boldsymbol{\theta}}(\mathbf{x}) = \int P_{\boldsymbol{\theta}}(\mathbf{x}, \boldsymbol{z})\delta\boldsymbol{z} = \int P_{\boldsymbol{\theta}}(\boldsymbol{z}|\mathbf{x})P_{\boldsymbol{\theta}}(\mathbf{x})\delta\boldsymbol{z} = \int P_{\boldsymbol{\theta}}(\mathbf{x}|\boldsymbol{z})P(\boldsymbol{z})\delta\boldsymbol{z} \tag{1}$$

We model the conditional distribution $P_{\boldsymbol{\theta}}(\mathbf{x}|\boldsymbol{z})$ by a *decoder*, typically a neural network. Very often the marginal likelihood $P_{\boldsymbol{\theta}}(\mathbf{x})$ will be intractable because the integral in equation (1) does not have an analytical form nor an efficient estimator (Kingma (2017)). As a result the respective posterior distribution, $P_{\boldsymbol{\theta}}(\boldsymbol{z}|\mathbf{x})$, is also intractable.

Variational inference side-steps the intractability of the posterior by approximating it with a tractable distribution $Q_{\boldsymbol{\phi}}(\boldsymbol{z}|\mathbf{x}) \approx P_{\boldsymbol{\theta}}(\boldsymbol{z}|\mathbf{x})$. VAEs use an *encoder* (generally a neural network) to model the approximate posterior $Q_{\boldsymbol{\phi}}(\boldsymbol{z}|\mathbf{x})$ and optimize the parameters $\boldsymbol{\phi}$ to minimize the reverse KL divergence $KL[Q_{\boldsymbol{\phi}}(\boldsymbol{z}|\mathbf{x})||P_{\boldsymbol{\theta}}(\boldsymbol{z}|\mathbf{x})]$ between the approximate posterior distribution $Q_{\boldsymbol{\phi}}(\boldsymbol{z}|\mathbf{x})$ and the true posterior $P_{\boldsymbol{\theta}}(\boldsymbol{z}|\mathbf{x})$. Given that $Q_{\boldsymbol{\phi}}(\boldsymbol{z}|\mathbf{x})$ is a powerful model (such that the KL divergence against the true posterior will be close to zero) we maximize the tractable Evidence Lower BOund (ELBO) to the intractable marginal likelihood. $\mathcal{L}_{\boldsymbol{\theta}}(\mathbf{x}) \leq P_{\boldsymbol{\theta}}(\mathbf{x})$ (full derivation available in the appendix)

$$\text{ELBO:} \quad \mathcal{L}_{\boldsymbol{\theta}}(\mathbf{x}) = \mathbb{E}_{Q_{\boldsymbol{\phi}}(\boldsymbol{z}|\mathbf{x})}[\log P_{\boldsymbol{\theta}}(\mathbf{x}|\boldsymbol{z})] - KL[Q_{\boldsymbol{\phi}}(\boldsymbol{z}|\mathbf{x}) \,||\, P(\boldsymbol{z})] \tag{2}$$

By sharing the variational parameters $\boldsymbol{\phi}$ of the encoder across the data points (*amortized inference* Gershman & Goodman (2014)), variational autoencoders avoid per-data optimization loops typically needed by mean-field approaches.

### 3.1 Sequential Generative Modeling

The standard setting in maximum-likelihood generative modeling is to estimate the set of parameters $\boldsymbol{\theta}$ that will maximize the marginal likelihood $P_{\boldsymbol{\theta}}(\mathbf{x})$ for data sample $\mathbf{X}$ generated IID from a single true data distribution $P^*(\mathbf{x})$. In our work we assume the data are generated from multiple distributions $P_i^*(\mathbf{x})$ such that $P^*(\mathbf{x}) = \sum_i \pi_i^* P_i^*(\mathbf{x})$. In classical batch generative modelling, the individual data points are not associated with the specific generative distributions $P_i^*(\mathbf{x})$. Instead, the whole sample $\mathbf{X}$ is considered to be generated from the mixture distribution $P^*(\mathbf{x})$. Latent variable models $P_{\boldsymbol{\theta}}(\mathbf{x}, \mathbf{z}) = P_{\boldsymbol{\theta}}(\mathbf{x}|\mathbf{z})P(\mathbf{z})$ (such as VAEs) capture the complex structures in $P^*(\mathbf{x})$ by conditioning the observed variables $\mathbf{x}$ on the latent variables $\mathbf{z}$ and combining these in (possibly infinite) mixtures $P_{\boldsymbol{\theta}}(\mathbf{x}) = \int P_{\boldsymbol{\theta}}(\mathbf{x}|\mathbf{z})P(\mathbf{z})\delta\mathbf{z}$.

Our sequential setting is vastly different from the batch approach described above. We receive a stream of (possibly infinite) data $\mathbf{X} = \{\mathbf{X}_1, \mathbf{X}_2, \ldots\}$ where the data samples $\mathbf{X}_i = \{\mathbf{x}_i^{(1)}, \mathbf{x}_i^{(2)}, \ldots, \mathbf{x}_i^{(K_i)}\}$ originate from the components $P_i^*(\mathbf{x})$ of the generative distribution. At any given time we observe the latest sample $\mathbf{X}_i$ generated from a single component $P_i^*(\mathbf{x})$ without access to any of the previous samples generated by the other components of $P^*(\mathbf{x})$. Our goal is to sequentially build an approximation $P_{\boldsymbol{\theta}}(\mathbf{x})$ of the true mixture $P^*(\mathbf{x})$ by only observing data from a single component $P_i^*(\mathbf{x})$ at a time.

## 4 Model

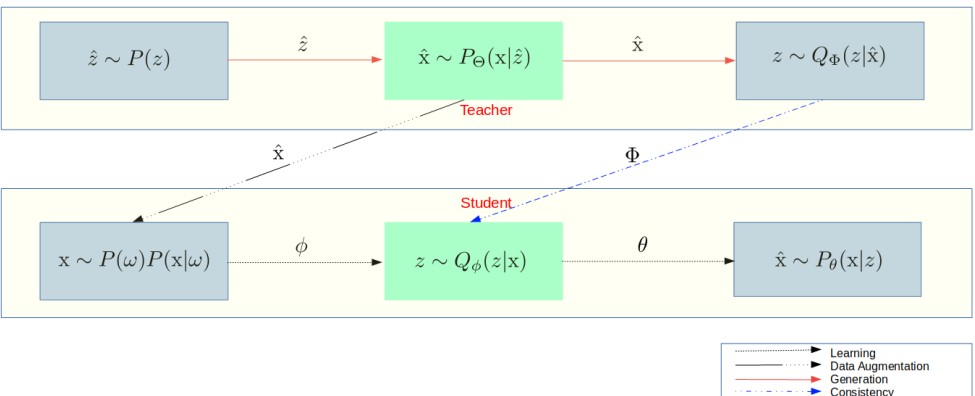

Figure 2: Shown above is the relationship of the teacher and the student generative models. Data generated from the teacher model is used to augment the student model's training data and consistency is applied between posteriors. Best viewed in color.

To enable lifelong generative learning we propose a dual model architecture based on a student-teacher model. The teacher and the student have rather different roles throughout the learning process: the teacher's role is to preserve the memory of the previously learned tasks and to pass this knowledge onto the student; the student's role is to learn the distributions over the new incoming data while accommodating for the knowledge obtained from the teacher. The dual model architecture is summarized in figure 2.

The top part represents the teacher model. At any given time the teacher contains a summary of all previous distributions within the learned parameters of the encoder $Q_{\boldsymbol{\Phi}}(\boldsymbol{z}|\mathbf{x})$ and the decoder $P_{\boldsymbol{\Theta}}(\mathbf{x}|\boldsymbol{z})$. The teacher is used to generate synthetic samples $\hat{\mathbf{x}}$ from these past distributions by decoding samples from the prior $\hat{\boldsymbol{z}} \sim P(\boldsymbol{z})$ through the decoder $\hat{\mathbf{x}} \sim P_{\boldsymbol{\Theta}}(\mathbf{x}|\hat{\boldsymbol{z}})$. The generated synthetic samples $\hat{\mathbf{x}}$ are passed onto the student model as a form of knowledge transfer about the past distributions.

The bottom part of figure 2 represents the student, which is responsible for updating the parameters of the encoder $Q_{\boldsymbol{\phi}}(\boldsymbol{z}|\mathbf{x})$ and decoder $P_{\boldsymbol{\theta}}(\mathbf{x}|\boldsymbol{z})$ models over the newly observed data. The student is exposed to a mixture of learning instances $\mathbf{x}$ sampled from $\mathbf{x} \sim P(\boldsymbol{\omega})P(\mathbf{x}|\boldsymbol{\omega})$, $\boldsymbol{\omega} \sim \text{Ber}(\pi)$; it sees synthetic instances generated by the teacher $P(\mathbf{x}|\boldsymbol{\omega} = 0) = P_{\boldsymbol{\Theta}}(\mathbf{x}|\boldsymbol{z})$, and real ones sampled from the currently active training distribution $P(\mathbf{x}|\boldsymbol{\omega} = 1) = P^*(\mathbf{x})$. The mean $\pi$ of the Bernouli distribution controls the sampling proportion of the previously learnt distributions to the current one.

If we have seen k distinct distributions prior to the currently active one then $\pi = \frac{k}{k+1}$. In this way we ensure that all the past distributions and the current one are equally represented in the training set used by the student model.

Once a new distribution is signalled, the old teacher is dropped, the student model is frozen and becomes the new teacher ($\phi \rightarrow \Phi, \theta \rightarrow \Theta$), and a new student is initiated with the latest weights $\phi$ and $\theta$ from the previous student (the new teacher).

### 4.1 TEACHER-STUDENT CONSISTENCY

Each new student instantiation uses the input data mix to learn a new approximate posterior $Q_\phi(z|\mathbf{x})$. In addition to being initiated by the new teacher's weights and receiving information about the teacher's knowledge via the synthetic samples $\hat{\mathbf{x}}$, we further foster the lifelong learning idea by bringing the latent variable posterior induced by the student model closer to the respective posterior induced by the teacher model. We enforce the latter constraint *only over the synthetic samples*, ensuring that the previously learnt latent variable posteriors are preserved over the different models. In doing so, we alleviate the effect of catastrophic interference.

To achieve this, we complement the classical VAE objective (equation (2)) with a term minimizing the KL divergence $KL[Q_\phi(z|\hat{\mathbf{x}})||Q_\Phi(z|\hat{\mathbf{x}})]$ between the student's and the teacher's posteriors over the synthetic data $\hat{\mathbf{x}}$. The teacher's encoder model, which already has the accumulated knowledge from the previous learning steps, is thus reused within the new student's objective. Under certain mild assumptions, we show that this objective reparameterizes the student model's posterior, while preserving the same learning objective as a standard VAE (appendix section 7.0.1).

### 4.2 LATENT VARIABLE

A critical component of our model is the synthetic data generation by the teacher's decoder $\hat{\mathbf{x}} \sim P_\Theta(\mathbf{x}|z)$. The synthetic samples need to be representative of all the previously observed distributions in order to provide the student with ample information about the learning history. The teacher generates these synthetic samples by first sampling the latent variable from the prior $\hat{z} \sim P(z)$ followed by the decoding step $\hat{\mathbf{x}} \sim P_\Theta(\mathbf{x}|\hat{z})$. As we will describe shortly, the latent variable $\hat{z}$ has a categorical component which corresponds to all the past distributions. This categorical component allows us to uniformly sample synthetic instances from all past distributions.

A simple unimodal prior distribution $P(z)$, such as the isotropic Gaussian typically used in classical VAEs, results in an undersampling of the data points that are mapped to a posterior mean that is further away from the prior mean. Visualizing the 2d latent posterior of MNIST in figure 1(b) allows us to get a better intuition of this problem. If for example the prior mean corresponds to a point in latent space between two disparate distributions, the sample generated will not correspond to a sample from the real distribution. Since we use synthetic samples from the teacher in the student model, this aliased sample corresponding to the prior mean, will be reused over and over again, causing corruption in the learning process. In addition, we would under represent the respective true distributions in the learning input mix of the student and eventually lead to distribution loss.

We circumvent this in our model by decomposing the latent variable $z$ into a discrete component $z_d \in \mathcal{R}^J$ and a continuous component $z_c \in \mathcal{R}^F$, $z = [z_d, z_c]$. The discrete component $z_d$ shall summarise the most discriminative information about each of the true generating distributions $P_i^*(\mathbf{x})$. We use the uniform multivariate categorical prior $z_d \sim Cat(\frac{1}{J})$ to represent it and the same parametric family for the approximate posterior $Q_\Phi(z|\mathbf{x})$. The continuous $z_c$ component is the global representation of the distributional variability and we use the multivariate standard normal as the prior $z_c \sim N(\mathbf{0}, \mathbf{I})$ and the isotropic multivariate normal $N(\boldsymbol{\mu}, \sigma^2 \mathbf{I})$ for the approximate posterior.

When generating synthetic data, the teacher now independently samples from the discrete and continuous priors $\hat{z}_d \sim P(z_d), \hat{z}_c \sim P(z_c)$ and uses the composition of these to condition the decoding step $\hat{\mathbf{x}} \sim P_\Theta(\mathbf{x}|\hat{z}_d, \hat{z}_c)$. Since the discrete representation $\hat{z}_d$ is associated with the true generative distribution components $P_i^*(\mathbf{x})$, uniformly sampling the discrete prior ensures that that the distributions are well represented in the synthetic mix that the student observes.

In general, the capacity of a categorical distribution is less than that of a continuous normal distribution. To prevent the VAE's encoder from using primarily the continuous representation while disregarding the discrete one we further complement the learning objective by a term maximising the mutual information between the discrete representation and the data $I(z_d; \mathbf{x}) = H(z_d) - H(z_d|\mathbf{x})$. $H(z_d)$ is used to denote the marginal entropy of $z_d$ and $H(z_d|\mathbf{x})$ denotes the conditional entropy of $z_d$ given $\mathbf{x}$.[1]

## 4.3 LEARNING OBJECTIVE

The final learning objective for each of the student models is the maximization of the ELBO from equation (2), augmented by the negative of the cross-model consistency term introduced in section 4.1 and the mutual information term proposed in section 4.2.

$$\underbrace{\mathbb{E}_{Q_\phi}[log\, P_\theta(\mathbf{x}|z)] - KL[Q_\phi(z|\mathbf{x})||P(z)]}_{\text{VAE ELBO}} - \underbrace{\mathbb{1}(\boldsymbol{\omega}=0)KL[Q_\phi(z_d|\mathbf{x})||Q_\Phi(z_d|\mathbf{x})]}_{\text{Consistency Regularizer}} + \underbrace{\lambda I(z_d;\mathbf{x})}_{\text{Mutual Info}} \;,$$

(3)

We sample the training instances $\mathbf{x}$ from $\mathbf{x} \sim P(\boldsymbol{\omega})P(\mathbf{x}|\boldsymbol{\omega}), \boldsymbol{\omega} \sim Ber(\pi)$ as described in section 4. Thus they can either be generated from the teacher model ($\boldsymbol{\omega} = 0$) or come from the training set of the currently active distribution ($\boldsymbol{\omega} = 1$). $\mathbb{1}(.)$ is the indicator function which evaluates to 1 if its argument is true and zero otherwise; it makes sure that the consistency regularizer is applied only over the synthetic samples generated by the teacher. The $\lambda$ hyper-parameter controls the importance of the mutual information regularizer. We present the analytical evaluation of the consistency regularizer in appendix section 7.0.1.

## 5 EXPERIMENTS

We conducted a set of experiments to explore the behaviour and properties of the method we propose. We specifically concentrate on the benefits our model brings in the lifelong learning setting which is the main motivation of our work. We explain the settings of the individual experiments and their focus in the following three sections.

In all the experiments we use the notion of a distributional 'interval': the interval in which we observe samples from a single distribution $P_i^*(\mathbf{x})$ before the transition to the next distribution $P_{i+1}^*(\mathbf{x})$ occurs. The length of the intervals is in principle random and we developed a heuristic to generate these. We provide further details on this together with other technical details related to the network implementation and training common for all the experiments in the appendix.

### 5.1 FASHION MNIST : SEQUENTIAL GENERATION

In this experiment, we seek to establish the performance benefit that our augmented objective formulation in section 4.3 brings into the learning in contrast to the simple ELBO objective 2. We do so by training two models with identical student-teacher architectures as introduced in section 4, with one using the consistency and mutual information augmented objective (*with consistency*) and the other using the standard ELBO objective (*without consistency*). We also demonstrate the ability of our model to disambiguate distributional boundaries from the distributional variations.

We use Fashion MNIST Xiao et al. (2017) [2] to simulate our sequential learning setting. We treat each object as a different distribution and present the model with samples drawn from a single distribution at a time. We sequentially progress over the ten available distributions. When a distribution transition occurs (new object) we signal the model, make the latest student the new teacher and instantiate a new student model.

We quantify the performance of the generative models by computing the ELBO over the standard Fashion MNIST test set after every distributional transition. The test set contains objects from all of the individual distributions. We run this procedure ten times and report the average test ELBO over the ten repetitions in figure 3(c). We see that around the 3rd interval (the 3rd distributional

---

[1]A similar idea is leveraged in InfoGAN Chen et al. (2016).

[2]We do a similar experiment over MNIST in the appendix

Figure 3: (a) Generation *with consistency* regularizer. (b) Generation *without consistency* regularizer. (c) Average negative ELBO over ten trials (each) for the ten distributions within Fashion MNIST.

transition), the negative ELBO of the *with consistency* model is systematically below ($\sim 20$ nats ) that of the *without consistency* model. This confirms the benefits of our new objective formulation for reducing the effects of the catastrophic interference, a crucial property in our lifelong learning setting. In the same figure we also plot the ELBO of the baseline batch VAE. The batch VAE will always outperform our model because it has simultaneous access to all of the distributions during training.

After observing and training over all ten distributions we generate samples from the final students of the two models. We do this by fixing the discrete distribution $z_d$ to one-hot vectors over the whole categorical distribution, while randomly sampling the continuous prior $z_c \sim \mathcal{N}(0, I)$. We contrast samples generated from the model *with consistency* (figure 3(a)) to the model *without consistency* (figure 3(b)). Our model learns to separate 'style' from the distributional boundaries. For example, in the last row of our *with consistency* model, we observe the various styles of shoes. The *without consistency* model mixes the distributions randomly. This illustrates the benefits that our augmented objective has for achieving consistent sampling from the individual distributional components.

## 5.2 ROTATED MNIST : LONG TERM DISTRIBUTION ACCUMULATION

In this experiment we dig deeper into the benefits our objective formulation brings for the lifelong learning setting. We expose the models to a much larger number of distributions and we explore how our augmented objective from 4.3 helps in preserving the previously learned knowledge. As in section 5.1, we compare models *with* and *without* consistency with identical teacher-student architectures. We measure the ability of the models to recall the previously learned information by looking at the consistency between the posterior of the student and the teacher models over the test data set

$$\text{consistency:} \quad \#\{k : Q_{\Phi}(z_d|\mathbf{x}_k) == Q_{\phi}(z_d|\mathbf{x}_k), \mathbf{x}_k \in \mathbf{X}_{test}\} \ . \quad (4)$$

We use the MNIST dataset in which we rotate each of the original digit samples by angles $\nu = [30°, 70°, 130°, 200°, 250°]$. We treat each rotation of a single digit family as an individual distribution $\{P_i^*(\mathbf{x})\}_{i=1}^{70}$. Within each distributional interval, we sample the data by first sampling (uniformly with replacement) one of the 70 distributions and then sampling the data instances $\mathbf{x}$ from the selected distribution.

Figure 4(b) compares the consistency results of the two tested models throughout the learning process. Our model *with* the augmented objective clearly outperforms the model that uses the simple ELBO objective. This confirms the usefulness of the additional terms in our objective for preserving the previously learned knowledge in accordance with the lifelong learning paradigms. In addition, similarly as in experiment 5.1, figure 4(a) documents that the model *with* the augmented objective (thanks to reducing the effects of the catastrophic interference) achieves lower negative test ELBO systematically over the much longer course of learning ($\sim 30$ nats).

We also visualise in figure 4(c) how the accumulation of knowledge speeds up the learning process. For each distributional interval we plot the norms of the model gradients across the learning iterations. We observe that for later distributional intervals the curves become steeper much quicker, reducing the gradients and reaching (lower) steady states much faster then in the early learning stages. This suggests that the latter models are able to learn quicker in our proposed architecture.

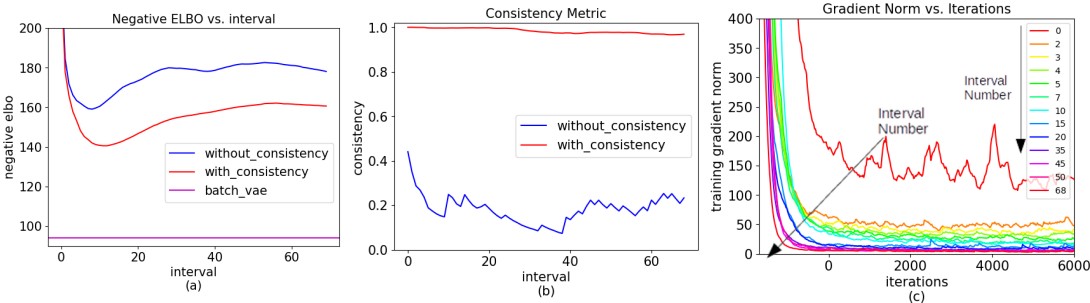

Figure 4: (a) Negative test ELBO over the learning history; (b) consistency between the teacher and student posteriors across the test data samples normalized by the test data set size; (c) speed of learning convergence across distributional intervals. Best viewed in color.

## 5.3 SVHN TO MNIST

In this experiment we explore the ability of our model to retain and transfer knowledge across completely different datasets. We use MNIST and SVHN Netzer et al. (2011) to demonstrate this. We treat all samples from SVHN as being generated by one distribution $P_1^*(\mathbf{x})$ and all the MNIST[3] samples as generated by another distribution $P_2^*(\mathbf{x})$ (irrespective of the specific digit).

We first train a student model (standard VAE) over the entire SVHN data set. Once done, we freeze the parameters of the encoder and the decoder and transfer the model into the teacher state ($\phi \to \Phi, \theta \to \Theta$). We then use this teacher to aid the learning of the new student over the mix of the teacher-generated synthetic SVHN samples $\hat{\mathbf{x}}$ and the true MNIST data.

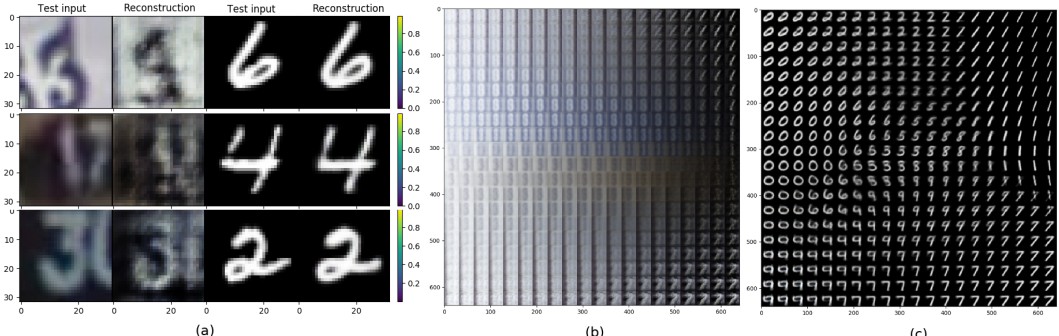

Figure 5: (a) Reconstructions of test samples from SVHN[left] and MNIST[right]; (b) Decoded samples $\hat{\mathbf{x}} \sim P_\theta(\mathbf{x}|\mathbf{z}_d, \mathbf{z}_c)$ based on linear interpolation of $\mathbf{z}_c \in \mathcal{R}^2$ with $\mathbf{z}_d = [0, 1]$; (c) Same as (b) but with $\mathbf{z}_d = [1, 0]$.

We use the final student model to reconstruct samples from the two datasets by passing them through the learned encoding/decoding flow: $\mathbf{x} \sim P_i^*(\mathbf{x}) \to \mathbf{z} \sim Q_\phi(\mathbf{z}|\mathbf{x}) \to \hat{\mathbf{x}} \sim P_\theta(\mathbf{x}|\mathbf{z})$. We visualise examples of the true inputs $\mathbf{x}$ and the respective reconstructions $\hat{\mathbf{x}}$ in figure 5(a). We see that even though the only true data the final model received for training were from MNIST, it can still reconstruct SVHN data. This confirms the ability of our architecture to transition between complex distributions while still preserving the knowledge learned from the previously observed distributions.

Finally, in figure 5(b) and 5(c) we illustrate the data generated from an interpolation of a 2-dimensional continuous latent space. For this we specifically trained the models with the continuous latent variable $\mathbf{z}_c \in \mathcal{R}^2$. To generate the data, we fix the discrete categorical $\mathbf{z}_d$ to one of the possible values $\{[0, 1], [1, 0]\}$ and linearly interpolate the continuous $\mathbf{z}_c$ over the range $[-3, 3]$. We then decode these to obtain the samples $\hat{\mathbf{x}} \sim P_\theta(\mathbf{x}|\mathbf{z}_d, \mathbf{z}_c)$. The model learns a common

---

[3]In order to work over both of these datasets we convert MNIST to RGB and resize it to 32x32 to make it consistent with the dimensions of SVHN.

continuous structure for the two distributions which can be followed by observing the development in the generated samples from top left to bottom right on both figure 5(b) and 5(c).

## 6 CONCLUSION

In this work we propose a novel method for learning generative models over streaming data following the lifelong learning principles. The principal assumption for the data is that they are generated by multiple distributions and presented to the learner in a sequential manner (a set of observations from a single distribution followed by a distributional transition). A key limitation for the learning is that the method can only access data generated by the current distribution and has no access to any of the data generated by any of the previous distributions.

The proposed method is based on a dual student-teacher architecture where the teacher's role is to preserve the past knowledge and aid the student in future learning. We argue for and augment the standard VAE's ELBO objective by terms helping the teacher-student knowledge transfer. We demonstrate on a series of experiments the benefits this augmented objective brings in the lifelong learning settings by supporting the retention of previously learned knowledge (models) and limiting the usual effects of catastrophic interference.

In our future work we will explore the possibilities to extend our architecture to GAN-like Goodfellow et al. (2014) learning with the prospect to further improve the generative abilities of our method. GANs, however, do not use a metric for measuring the quality of the learned distributions such as the marginal likelihood or the ELBO in their objective and therefore the transfer of our architecture to these is not straightforward.

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

## 7 APPENDIX

### 7.0.1 UNDERSTANDING THE CONSISTENCY REGULARIZER

The analytical derivations of the consistency regularizer show that the regularizer can be interpreted as an a transformation of the standard VAE regularizer. In the case of an isotropic gaussian posterior, the proposed regularizer scales the mean and variance of the student posterior by the variance of the teacher 7.0.2 and adds an extra 'volume' term. This interpretation of the consistency regularizer shows that the proposed regularizer preserves the same learning objective as that of the standard VAE. Below we present the analytical form of the consistency regularizer with categorical and isotropic gaussian posteriors:

**Corollary 7.0.1** *We parameterize the learnt posterior of the teacher by* $\Phi_i = \frac{\exp(p_i^E)}{\sum_{i=1}^{J}\exp(p_i^E)}$ *and the posterior of the student by* $\phi_i = \frac{\exp(p_i^S)}{\sum_{i=1}^{J}\exp(p_i^S)}$. *We also redefine the normalizing constants as* $c^E = \sum_{i=1}^{J}\exp(p_i^E)$ *and* $c^S = \sum_{i=1}^{J}\exp(p_i^S)$ *for the teacher and student models respectively. The reverse KL divergence in equation 8 can now be re-written as:*

$$KL(Q_\phi(\boldsymbol{z}_d|\boldsymbol{x})||Q_\Phi(\boldsymbol{z}_d|\boldsymbol{x})) = \sum_{i=1}^{J}\frac{\exp(p_i^S)}{c^S}log\left(\frac{\exp(p_i^S)}{c^S}\frac{c^E}{\exp(p_i^E)}\right) \quad (5)$$

$$= H(\boldsymbol{p}^S, \boldsymbol{p}^S - \boldsymbol{p}^E) = -H(\boldsymbol{p}^s) + H(\boldsymbol{p}^S, \boldsymbol{p}^E)$$

*where* $H(\_)$ *is the entropy operator and* $H(\_,\_)$ *is the cross-entropy operator.*

**Corollary 7.0.2** *We assume the learnt posterior of the teacher is parameterized by a centered, isotropic gaussian with* $\Phi = [\boldsymbol{\mu}^E = \boldsymbol{0}, \Sigma^E = \boldsymbol{\sigma}^{E^2}\boldsymbol{I}]$ *and the posterior of our student by a non-centered isotropic gaussian with* $\phi = [\boldsymbol{\mu}^S, \Sigma^S = \boldsymbol{\sigma}^{S2}\boldsymbol{I}]$, *then*

$$KL(Q_\phi(\boldsymbol{z}|\boldsymbol{x})||Q_\Phi(\boldsymbol{z}|\boldsymbol{x})) = 0.5\left[tr(\Sigma^{E^{-1}}\Sigma^S) + (\boldsymbol{\mu}^E - \boldsymbol{\mu}^S)^T\Sigma^{E^{-1}}(\boldsymbol{\mu}^E - \boldsymbol{\mu}^S) - F + log\left(\frac{|\Sigma^E|}{|\Sigma^S|}\right)\right]$$

$$= 0.5\sum_{j=1}^{F}\left[\frac{1}{\sigma^{E2}(j)}(\sigma^{S2}(j) + \mu^{S2}(j)) - 1 + log\ \sigma^{E2}(j) - log\ \sigma^{S2}(j)\right]$$

$$= KL(Q_{\phi^*}(\boldsymbol{z}|\boldsymbol{x})||\mathcal{N}(0, \boldsymbol{I})) - log\ |\Sigma^E|$$

$$(6)$$

*Via a reparameterization of the student's parameters:*

$$\phi^* = [\boldsymbol{\mu}^{S*}, \boldsymbol{\sigma}^{S*2}]$$

$$\boldsymbol{\mu}^{S*} = \frac{\mu^S(j)}{\sigma^{E2}(j)}; \boldsymbol{\sigma}^{S*2} = \frac{\sigma^{S2}(j)}{\sigma^{E2}(j)} \quad (7)$$

It is also interesting to note that our posterior regularizer becomes the prior if:

$$lim_{\boldsymbol{\sigma}^{E2}\mapsto 1}KL(Q_\phi(\boldsymbol{z}|\boldsymbol{x})||Q_\Phi(\boldsymbol{z}|\boldsymbol{x})) = KL(Q_\phi(\boldsymbol{z}|\boldsymbol{x})||\mathcal{N}(0, \boldsymbol{I}))$$

### 7.1 ELBO DERIVATION

Variational inference Hoffman et al. (2013) side-steps the intractability of the posterior distribution by approximating it with a tractable distribution $Q_\Phi(\boldsymbol{z}|\mathbf{x})$; we then optimize the parameters $\Phi$ in order to bring this distribution close to $P_\Phi(\boldsymbol{z}|\mathbf{x})$. The form of this approximate distribution is fixed and is generally conjugate to the prior $P(\boldsymbol{z})$. Variational inference converts the problem of posterior inference into an optimization problem over $\Phi$. This allows us to utilize stochastic gradient descent to solve our problem. To be more concrete, variational inference tries to minimize the reverse Kullback-Leibler (KL) divergence between the variational posterior distribution $Q_\Phi(\boldsymbol{z}|\mathbf{x})$ and the true posterior $P_\theta(\boldsymbol{z}|\mathbf{x})$:

$$KL[Q_{\mathbf{\Phi}}(\boldsymbol{z}|\mathbf{x})||P_{\boldsymbol{\theta}}(\boldsymbol{z}|\mathbf{x})] = \log P_{\boldsymbol{\theta}}(\mathbf{x}) - \underbrace{\mathbb{E}_{Q_{\mathbf{\Phi}}(\boldsymbol{z}|\mathbf{x})}\left[\log \frac{P_{\boldsymbol{\theta}}(x,z)}{Q_{\mathbf{\Phi}}(\boldsymbol{z}|\mathbf{x})}\right]}_{\mathcal{L}_{\boldsymbol{\theta}}} \tag{8}$$

Rearranging the terms in equation 8 and utilizing the fact that the KL divergence is a measure, we can derive the evidence lower bound $\mathcal{L}_{\boldsymbol{\theta}}$ (ELBO) which is the objective function we directly optimize:

$$log\, P_{\boldsymbol{\theta}}(\mathbf{x}) \geq \mathbb{E}_{Q_{\mathbf{\Phi}}(\boldsymbol{z}|\mathbf{x})}[log\, P_{\boldsymbol{\theta}}(\mathbf{x}|\boldsymbol{z})] - KL(Q_{\mathbf{\Phi}}(\boldsymbol{z}|\mathbf{x}) \,||\, P(\boldsymbol{z})) = \mathcal{L}_{\boldsymbol{\theta}} \tag{9}$$

In order to backpropagate it is necessary to remove the dependence on the stochastic variable $\boldsymbol{z}$. To achieve this, we push the sampling operation outside of the computational graph for the normal distribution via the reparameterization trick Kingma & Welling (2014) and the gumbel-softmax reparameterization Maddison et al. (2016); Jang et al. (2017) for the discrete distribution. In essence the reparameterization trick allows us to introduce a distribution $P(\epsilon)$ that is not a function of the data or computational graph in order to move the gradient operator into the expectation:

$$\nabla \, \mathbb{E}_{Q_{\mathbf{\Phi}}(\boldsymbol{z}|\mathbf{x})}\left[\log \frac{P_{\boldsymbol{\theta}}(x,z)}{Q_{\mathbf{\Phi}}(\boldsymbol{z}|\mathbf{x})}\right] \mapsto \mathbb{E}_{P(\epsilon)}\left[\nabla \log \frac{P_{\boldsymbol{\theta}}(x,z)}{Q_{\mathbf{\Phi}}(\boldsymbol{z}|\mathbf{x})}\right] \tag{10}$$

## 7.2 MODEL RELATED

In this section we provide extra details of our model architecture.

### 7.2.1 MODEL ARCHITECTURE

We utilized two different architectures for our experiments. The first two utilize a standard deep neural network with two layers of 512 to map to the latent representation and two layers of 512 to map back to the reconstruction for the decoder. We used batch norm Ioffe & Szegedy (2015) and ELU activations for all the layers barring the layer projecting into the latent representation and the output layer.

The final experiment with the transfer from SVHN to MNIST utilizes a fully convolutional architecture with only strided convolutional layers in the encoder (where the number of filters are doubled at each layer). The final projection layer for the encoder maps the data to a [C=$|z_d|$, 1, 1] output which is then reparameterized in the standard way. The decoder utilizes fractional strides for the convolutional-transpose (de-convolution) layers where we reduce the number of filters in half at each layer. The full architecture can be examined in our code repository [which will be de-anonymized after the review process]. All layers used batch norm Ioffe & Szegedy (2015) and ELU activations.

We utilized Adam Kingma & Ba (2015) to optimize all of our problems with a learning rate of 1e-4. When we utilized weight transfer we re-initialized the accumulated momentum vector of Adam as well as the aggregated mean and covariance of the Batch Norm layers. Our code is already available online under an MIT license at [4]

### 7.2.2 GUMBEL REPARAMETERIZATION

Since we model our latent variable as a combination of a discrete and a continuous distribution we also use the Gumbel-Softmax reparameterization Maddison et al. (2016); Jang et al. (2017). The Gumbel-Softmax reparameterization over logits [linear output of the last layer in the encoder] $\boldsymbol{p} \in \mathcal{R}^M$ and an annealed temperature parameter $\tau \in \mathcal{R}$ is defined as:

$$\boldsymbol{z} = softmax(\frac{log(\boldsymbol{p}) + \boldsymbol{g}}{\tau}); \boldsymbol{g} = -log(-log(\boldsymbol{u} \sim Unif(0,1))) \tag{11}$$

$\boldsymbol{u} \in \mathcal{R}^M, \boldsymbol{g} \in \mathcal{R}^M$. As the temperature parameter $\tau \mapsto 0$, $\boldsymbol{z}$ converges to a categorical.

---

[4]https://github.com/¡anonymized¿

### 7.2.3 EXPANDABLE MODEL CAPACITY AND REPRESENTATIONS

Multilayer neural networks with sigmoidal activations have a VC dimension bounded between $O(\rho^2)$Sontag (1998) and $O(\rho^4)$Karpinski & Macintyre (1997) where $\rho$ are the number of parameters. A model that is able to consistently add new information should also be able to expand its VC dimension by adding new parameters over time. Our formulation imposes no restrictions on the model architecture: i.e. new layers can be added freely to the new student model.

In addition we also allow the dimensionality of $z_d \in \mathcal{R}^J$, our discrete latent representation to grow in order to accommodate new distributions. This is possible because the KL divergence between two categorical distributions of different sizes can be evaluated by simply zero padding the teacher's smaller discrete distribution. Since we also transfer weights between the teacher and the student model, we need to handle the case of expanding latent representations appropriately. In the event that we add a new distribution we copy all the weights besides the ones immediately surrounding the projection into and out of the latent distribution. These surrounding weights are reinitialized to their standard Glorot initializations Glorot & Bengio (2010).

### 7.3 FORWARD VS. REVERSE KL

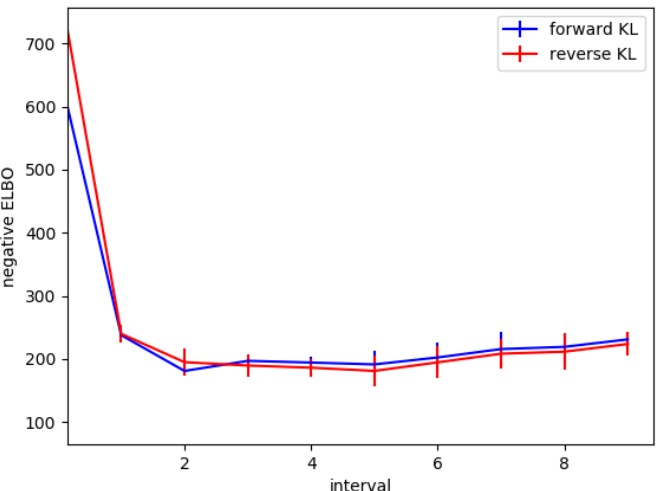

Figure 6: Reverse vs. forward KL on FashionMNIST 5.1

In our setting we have the ability to utilize the zero forcing (reverse or mode-seeking) KL or the zero avoiding (forward) KL divergence. In general, if the true underlying posterior is multi-modal, it is preferable to operate with the reverse KL divergence (Murphy (2012) 21.2.2). In addition, utilizing the mode-seeking KL divergence generates more realistic results when operating over image data.

In order to validate this, we repeat the experiment in 5.1. We train two models: one with the forward KL posterior regularizer and one with the reverse. We evaluate the -ELBO mean and variance over ten trials. Empirically, we observed no difference between the different measures. This is demonstrated in figure 6.

### 7.4 NUMBER OF REQUIRED SAMPLES

Our method derives its sample complexity from standard VAEs. In practice we evaluate the number of required real and synthetic samples by utilizing early stopping. When the negative ELBO on the validation set stops decreasing for 50 steps we stop training the current model and transition to the next distribution interval. Using this and the fact that we keep equal proportions of all observed distributions in our minibatch, we can evaluate the number of synthetic and real samples used during the single distribution interval. We demonstrate this procedure on experiment 5.1 in figure 7.

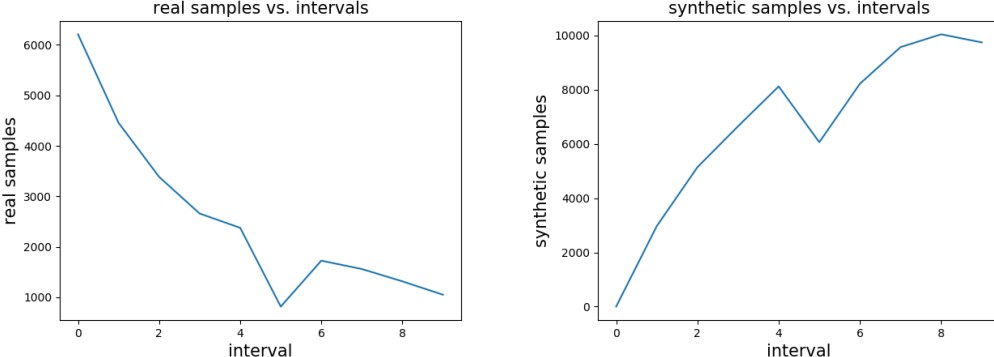

Figure 7: FashionMNIST experiment 5.1 data efficiency analysis. Left: Real samples used; Right: Synthetic samples used

We observe a rapid decrease of the number of required real samples as we assimilate more distributions into our model.

## 7.5 EXPERIMENTS RELATED

In this section we provide an extra experiment run on MNIST as well as some extra images from the rotated MNIST experiment.

### 7.5.1 MNIST : GENERATION AND ELBO

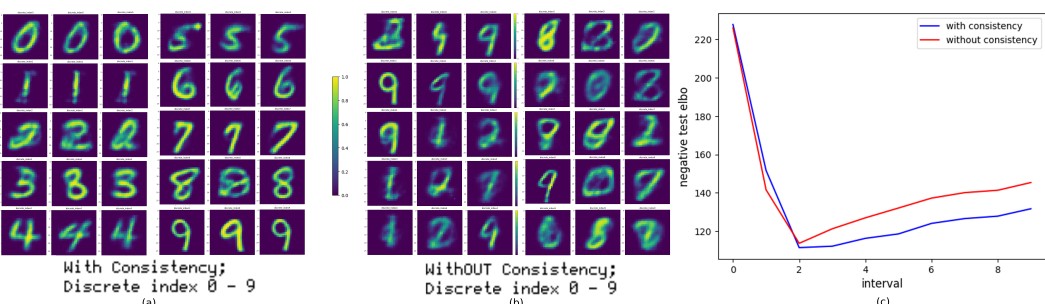

Figure 8: (a) Generation with consistency regularizer. (b) Without consistency regularizer. (c) Average log-likelihood over ten trials (each) for the ten separated distributions within MNIST.

In this experiment, we seek to establish the performance benefit that the consistency regularizer brings into the learning process. We do so by evaluating the ELBO for a model with and without the consistency and mutual information regularizers. We also demonstrate the ability of the regularizers to disambiguate distributional boundaries and their inter-distributional variations. I.e. for MNIST this separates the MNIST digits from their inter-class variants (i.e drawing style).

We use MNIST to simulate our sequential learning setting. We treat each digit as a different distribution and present the model with samples drawn from a single distribution at a time. For the purpose of this experiment we sequentially progress over the ten distributions (i.e. interval sampling involves linearly iterating over all the distributions ).

When an interval transition occurs we signal the model, make the student the new teacher and instantiate a new student model. We contrast this to a model that utilizes the same graphical model, without our consistency and mutual information regularizers. We quantify the performance of the generative models by computing the ELBO over the standard MNIST test set at every interval. The test set contains digits from all of the individual distributions. We run this procedure ten times and report the average ELBO over the test set.

After observing all ten distributions we evaluate samples generated from the final student model. We do this by fixing the discrete distribution $z_d$, while randomly sampling $z_c \sim \mathcal{N}(0, I)$. We contrast samples generated from the model with both regularizers (left-most image in 8) to the model without the regularizers (center image in 8). Our model learns to separate 'style' from distributional boundaries. This is demonstrated by observing the digit '2': i.e. different samples of $z_c$ produce different styles of writing a '2'.

### 7.5.2 ROTATED MNIST EXPERIMENT

We provide a larger sized image for the ELBO from experiment 5.2. We also visualize reconstructions from the rotated MNIST problem (visualized in figure 10). Finally in figure 11 we show the effects on the reconstructions when we do **not** use the mutual information regularizer. We believe this is due to the fact that the network utilizes the larger continuous representation to model the discriminative aspects of the observed distribution.

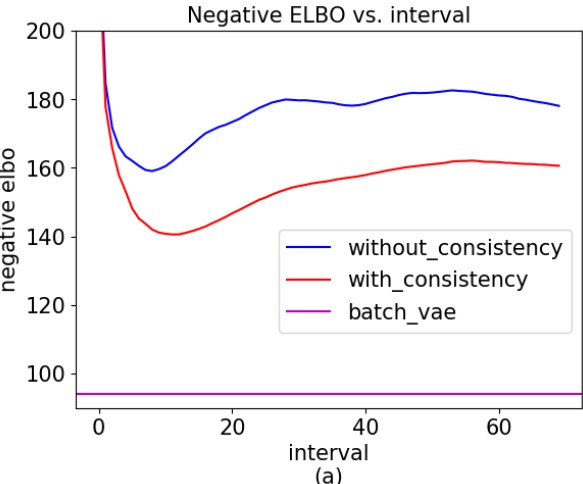

Figure 9: Visualization of ELBO for rotated MNIST evaluated at the last model (the one at the 70th interval)

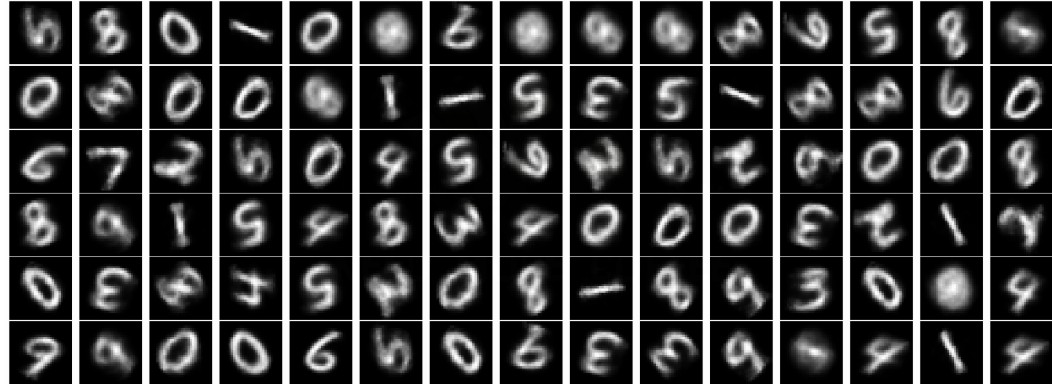

Figure 10: Visualization of reconstructions for rotated MNIST evaluated at the last model (the one at the 70th interval)

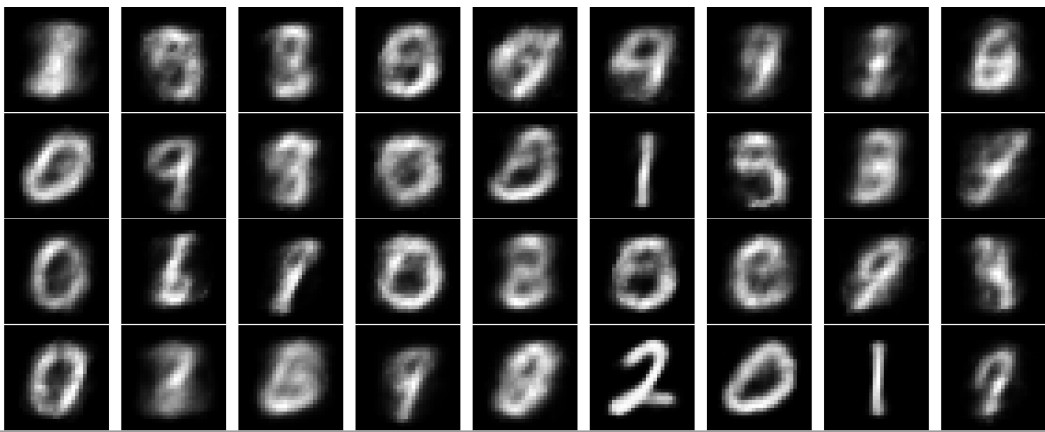

Figure 11: Visualization of reconstructions when we do not use the mutual information regularizer

