# OpenReview forum: "Lifelong Generative Modeling"
_ICLR.cc/2018/Conference — Reject_

### Official Review · AnonReviewer1 · 2017-11-18
**At last, a deep generative model addressing a fresh problem**

**Rating:** 9
**Confidence:** 5

**Review:**

We have seen numerous variants of variational autoencoders, most of them introducing delta changes to the original architecture to address the same sort of modeling problems. This paper attacks a different kind of problem, namely lifelong learning. This key aspect of the paper, besides the fact that it constitutes a very important problem, does also addes a strong element of freshness to the paper.

The construction of the generative model is correct, and commensurate with standard practice in the field of deep generative models. The derivations are correct, while the experimental evaluation is diverse and convincing.

---

> ### Author Response · Authors · 2017-12-11
> **Thanks for the feedback!**
>
> Thanks for your review of our paper! We try to address a problem we believe is novel and uncharted for generative models. Lifelong learning is a crucial part of transcending machine learning to make it useful in a more general environment. We appreciate your time in noticing the drastically disparate setting that we are operating over!

---

### Official Review · AnonReviewer2 · 2017-11-28
**Good initiative for using VAEs in a new framework. The work needs to be a bit more principled though.**

**Rating:** 4
**Confidence:** 5

**Review:**

- Second paragraph in Section 1: Nice motivation. I am not sure though whether the performed experiments are the most expressive for such motivation. For instance, is the experiment in Section 5.1 a common task in that sequential lifelong learning setting?

- Section 4, which is the main technical section of the paper, is quite full of lengthy descriptions that are a bit equivocal. I reckon each claim really needs to be supported by a corresponding unequivocal mathermatical formulation.

- An example of the last point can be found in Section 4.2: "The synthetic samples need to be representative of all the previously observed distributions ...": It will be much clearer how such samples are representative if a formulation follows, and that did not happen in Section 4.2.

- "1) Sampling the prior can select a point in the latent space that is in between two separate distributions ...": I am not sure I got this drawback of using the standard form of VAEs. Could you please further elaborate on this?

- "we restrict the posterior representation of the student model to **be close to that of the teacher** for the previous distributions** accumulated by the teacher. This allows the model parameters to **vary as necessary** in order to best fit the data": What if the previous distributions are not that close to the new one?

- Distribution intervals: Will it be the case in reality that these intervals will be given? Otherwise, what are the solutions to that? Can they be estimated somehow (as a future work)?


Minor:
- "we observe a sample X of K": sample X of size K, I guess?
- "... form nor an efficient estimator Kingma (2017)": citation style.
- "we illustrates ..."

---

> ### Author Response · Authors · 2017-12-11
> **Response to AnonReviewer2**
>
> Hi thanks  for your feedback! We hope we can address some of your comments below:
>
> Experiments: The experiments we do are standard experiments in a continual/lifelong setting [1,4] and catastrophic interference [2,3] and are adapted for the generative setting that we address in this paper. We used FashionMNIST to add more diversity in our experiments (we have a similar experiment as 5.1 for MNIST in appendix section 7.5)
>
> Representative Sampling & Formulations: in the second to last paragraph of section 4 (and in section 4.2) we do provide a formal description of how to sample the previous distributions represented in the training set of the student. However, we have added some additional text in both sections to make sure that there is no ambiguity. Briefly, we control the representativeness of the samples through the bernoulli distribution described in detail in section 4 (second to last paragraph). The mean of this distribution controls how many samples we sample from the previously learnt distributions vs. the current one. The previous distributions are uniformly sampled through the discrete latent variable of the teacher model which contains the most pertinent information about these distributions (section 4.2). We also provide additional mathematical formulation, such as a more theoretical understanding of the consistency regularizer, in the appendix section 7.0.1
>
> Prior sampling: for the following let us consider MNIST and its latent space representation given in figure 1. In a standard VAE the point corresponding to the mean of the prior  might be mapped by the encoder to a point in latent space that is in between a '9' and a '7'. This will generate an image that does not correspond to a real image. While in the standard VAE this might be ok or desirable, in the lifelong setting, repeating this operation over and over will cause corruption of the true underlying distribution. Since this point will also be sampled far more often, the model will not disambiguate between the '9' and the '7' distributions causing aliasing over time. This is a core issue that needs to be addressed in order to bring VAEs into the life-long setting. We do so through the introduction of the discrete component in the latent variable representation which allows us to disambiguate the true underlying distribution from its variability. This allows us to sample from any of the distributions seen so far without the aliasing problem described above.
>
>
> Varying posteriors: We believe that this is an important misunderstanding due to the way we had written equation 3; we have re- written this to clear up any confusion. Vanilla VAEs model the posterior distribution of the latent variables of any given instance as a normal distribution of which the mean and the variance are parameterized by the learned encoder network. Our consistency regulariser is applied {\em only} over the synthetic instances that are generated from the teacher model. It constrains the posterior distributions of their latent variables (as these are induced by the encoder of the teacher model) to be close to the respective posterior distributions induced by the encoder of the student model. The consistency regulariser {\em is not} applied to the instances of the new task. We make no statement and impose no constraint what so ever on the posterior induced by the student encoder on the instances of the {\em new} task.
>
> Distribution intervals: this is definitely something we considered, however in order to disambiguate the issues of inaccurate anomaly detection from the core problem, we decided to focus on the setting where both of these are provided to us. In future work we will attempt to develop a method that can simultaneously detect a distribution shift and model it into our framework.
>
> [1] Friedemann Zenke, Ben Poole, and Surya Ganguli. Continual learning through synaptic intelligence. In International Conference on Machine Learning, 2017.
>
> [2] Ian J. Goodfellow, Mehdi Mirza, Da Xiao, Aaron Courville, and Yoshua Bengio. An empirical investigation of catastrophic forgetting in gradient-based neural networks. In International Conference on Learning Representations, 2014a.
>
> [3] James Kirkpatrick, Razvan Pascanu, Neil Rabinowitz, Joel Veness, Guillaume Desjardins, Andrei A Rusu, Kieran Milan, John Quan, Tiago Ramalho, Agnieszka Grabska-Barwinska, et al. Overcoming catastrophic forgetting in neural networks. Proceedings of the National Academy of Sciences, pp. 201611835, 2017.
>
> [4] Lopez-Paz, David. "Gradient Episodic Memory for Continual Learning." Advances in Neural Information Processing Systems. 2017.

---

### Official Review · AnonReviewer3 · 2017-11-28
**Adapted VAE training to streaming data setting. Experiment shows the learned model can generate reasonable-looking samples. Unsure about quantitative results.**

**Rating:** 4
**Confidence:** 2

**Review:**

The paper proposed a teacher-student framework and a modified objective function to adapt VAE training to streaming data setting. The qualitative experimental result shows that the learned model can generate reasonable-looking samples. I'm not sure about what conclusion to make from the numerical result, as the test negative ELBO actually increased after decreasing initially. Why did it increase?

The modified objective function is a little ad-hoc, and it's unclear how to relate the overall objective function to Bayesian posterior inference (what exactly is the posterior that the encoder tries to approximate?). There is a term in the objective function that is synthetic data specific. Does that imply that the objective function is different depending on if the data is synthetic or real? What is the motivation/justification of choosing KL(Q_student||Q_teacher) as regularisation instead of the other way around? Would that make a difference in the goodness of the learned model? If not, wouldn't KL(Q_teacher||Q_student) result reduction in the variance of gradients and therefore a better choice?

Details on the minimum number of real samples per interval for the model to be able to learn is also missing. Also, how many synthetic samples per real samples are needed? How is the update with respect to synthetic sample scheduled? Given infinite amount of streaming data with a fixed number of classes/underlying distributions and interval length, and sample the class of each interval (uniformly) randomly, will the model/algorithm converge? Is there a minimum number of real examples that the student learner needs to see before it can be turned into a teacher?

Other question: How is the number of latent category J of the latent discrete distribution chosen?

Quality: The numerical experiment doesn't really compare to any other streaming benchmark and is a little unsatisfying. Without a streaming benchmark or a realistic motivating example in which the proposed scheme makes a significant difference, it's difficult to judge the contribution of this work.
Clarity: The manuscript is reasonably well-written. (minor: Paragraph 2, section 5, 'in principle' instead of 'in principal')
Originality: Average. The student-teacher framework by itself isn't novel. The modifications to the objective function appears to be novel as far as I am aware, but it doesn't require much special insights.
Significance: Below average. I think it will be very helpful if the authors can include a realistic motivating example where lifelong unsupervised learning is critical, and demonstrate that the proposed scheme makes a difference in the example.

---

> ### Author Response · Authors · 2017-12-11
> **Response to detailed feedback of AnonReviewer3 (part 1 of 2)**
>
> Thanks for your detailed review! Unsupervised learning is one of the most important challenge in machine learning; bringing it to a life-long setting is a crucial step towards systems that can continuously adapt their models of the world without supervision and without forgetting. We try to demonstrate some of the critical issues faced in transitioning VAE's to this setting and demonstrate an algorithm that allows for learning over long distributional intervals without access to any of the prior data. We would like to try to address some of your points below (we have a second part that address the rest of your comments as it didn't fit in one message) :
>
> ELBO Increase: In experiment 1 we present the model with data from a given distribution only within a single interval; the model never sees data from the same distribution again. Nevertheless we require it to reconstruct data points coming from all the distributions seen so far. As the number of uniquely seen distributions increases the task of reconstruction becomes more and more difficult which is why the -ELBO increases. In experiment 2 where the model might see a distribution again (due to sampling with replacement) we do observe the –ELBO decreasing.
>
> Bayesian posterior inference: Our goal with this work is to learn how to represent the mixture data distribution through time while only observing a single component at each interval. Our posterior comes into play because VAE's learn an approximate data distribution through the assumption of a latent variable model and the optimization of the ELBO objective. In a standard VAE, the learnt posterior is close to the prior. This is enforced through the KL divergence term of the standard ELBO. Similar to standard VAEs our learnt posterior is also close to the prior, but in addition we keep the inferred student posterior over the synthetic data close to their respective posterior inferred by the teacher. This ensures that the student’s encoder maps data samples to a similar latent space as the teacher (in order not to forget what the teacher has learned). Indeed as the reviewer notes, the objective function treats real data from the currently observable distribution in a different manner than the synthetic data; we do **not** constrain the posterior of data from the currently observed distribution to be similar to that of the teacher posterior. We have rewritten equation 3 to ensure that this point is clear. In the case of a VAE with an isotropic gaussian posterior, the consistency regularizer can be interpret as the standard VAE KL regularizer, with the mean and variance of the student posterior scaled by the variance of the teacher. We go over this in detail in appendix section 7.0.1.

---

> ### Author Response · Authors · 2017-12-11
> **Response to detailed feedback of AnonReviewer3 (part 2 of 2)**
>
> Forward vs. reverse KL: As the reviewer mentions the forward KL divergence does in fact provide a lower variance in theory, however since the underlying true posterior is generally multi-modal (and since we are working with images) it is preferable to work with the mode-seeking (i.e. reverse) KL divergence [2] in order to generate more realistic looking images. We have attached appendix section 7.3 with figures demonstrating that empirically there is almost no difference between both the measures.
>
> Minimum number of samples: Our model derives its sample complexity from standard VAEs. At each learning distributional interval we train our model on a new distribution (all the while ensuring not to forget previously learnt distributions) using early stopping. More precisely, when the negative ELBO on the validation set stops decreasing for 50 steps we consider training to be completed for that interval. We can then move to the next training set/distribution. Within each learning interval we make sure that synthetic instances from all past distributions as well as real instances from the current distribution are seen in the same proportion. We plot the number of learning instances (real and synthetic) seen at each learning episode until the stopping criterion is satisfied. We notice a rapid decrease in the number of real samples needed for learning as the number of observed distributions increases. We have added these graphs in section 7.4 of the appendix.
>
> Details on J: The dimension of the latent categorical J is grown over time. When we see the first distribution J=1; once there is a distribution transition (to an unobserved distribution) we set J=2.  We discuss this in a little more detail in section 7.2.3 of the appendix.
>
> Streaming benchmarks:  We think that there is a misunderstanding here. Our setting is not online/streaming generative modelling but continuous/life-long generative modelling. Online/streaming modelling seeks to update the model as instances arrive; if there is a distribution shift the model will adapt/shift to represent the current distribution and will forget previously learnt distributions. Instead in continual/life-long learning we do not want to forget the previously learned models because we might need to re-use them in the future as they can make learning of future distributions easier. The benchmarks we used are standard continual/ life-long learning benchmarks [3,6] and catastrophic interference benchmarks [4,5]. The comparison to online learning methods does not make sense and in fact will not be fair to these algorithms, since they do not try to retain all learned distributions; their performance will deteriorate rapidly as we see more and more distributions.
>
> References:
>
> [2] Murphy, Kevin P. Machine learning: a probabilistic perspective. MIT press, 2012. pp 733-734
>
> [3] Friedemann Zenke, Ben Poole, and Surya Ganguli. Continual learning through synaptic intelligence. In International Conference on Machine Learning, 2017.
>
> [4] Ian J. Goodfellow, Mehdi Mirza, Da Xiao, Aaron Courville, and Yoshua Bengio. An empirical investigation of catastrophic forgetting in gradient-based neural networks. In International Conference on Learning Representations, 2014a.
>
> [5] James Kirkpatrick, Razvan Pascanu, Neil Rabinowitz, Joel Veness, Guillaume Desjardins, Andrei A Rusu, Kieran Milan, John Quan, Tiago Ramalho, Agnieszka Grabska-Barwinska, et al. Overcoming catastrophic forgetting in neural networks. Proceedings of the National Academy of Sciences, pp. 201611835, 2017.
>
> [6] Lopez-Paz, David. "Gradient Episodic Memory for Continual Learning." Advances in Neural Information Processing Systems. 2017.

---

### Decision · Program_Chairs · 2018-01-29
**ICLR 2018 Conference Acceptance Decision**

**Decision:**

Reject

**Comment:**

Thank you for submitting you paper to ICLR. The paper studies an interesting problem and the solution, which fuses student-teacher approaches to continual learning and variational auto-encoders, is interesting. The revision of the paper has improved readability. However, although the framework is flexible, it is complex and appears rather ad hoc as currently presented. Exploration of the effect of the many hyper-parameters or some more supporting theoretical work / justification would help. The experimental comparisons were varied, but adding more baselines e.g. comparing to a parameter regularisation approach like EWC or synaptic intelligence applied to a standard VAE would have been enlightening.

Summary: There is the basis of a good paper here, but a comprehensive experimental evaluation of design choices or supporting theory would be useful for assessing what is a complex approach.